# To Vaccinate or Not: Impact of Bovine Viral Diarrhoea in French Cow-Calf Herds

**DOI:** 10.3390/vaccines9101137

**Published:** 2021-10-06

**Authors:** Sandie Arnoux, Fabrice Bidan, Alix Damman, Etienne Petit, Sébastien Assié, Pauline Ezanno

**Affiliations:** 1INRAE, Oniris, BIOEPAR, 44300 Nantes, France; sandie.arnoux@inrae.fr (S.A.); alix.damman@gmail.com (A.D.); sebastien.assie@oniris-nantes.fr (S.A.); 2Institut de L’élevage, 42 rue Georges Morel, F-49070 Beaucouzé, France; fabrice.bidan@idele.fr; 3GDS BFC, 21110 Breteniere, France; etienne.petit.gdsbfc@reseaugds.com

**Keywords:** epidemiology, model, endemic disease, control strategy, vaccination, economic impact, cattle, BVD

## Abstract

Bovine viral diarrhoea (BVD) remains an issue despite control programs implemented worldwide. Virus introduction can occur through contacts with neighbouring herds. Vaccination can locally protect exposed herds. However, virus spread depends on herd characteristics, which may impair vaccination efficiency. Using a within-herd epidemiological model, we compared three French cow-calf farming systems named by their main breed: Charolaise, Limousine, and Blonde d’Aquitaine. We assessed vaccination strategies of breeding females assuming two possible protections: against infection or against vertical transmission. Four commercial vaccines were considered: Bovilis^®^, Bovela^®^, Rispoval^®^, and Mucosiffa^®^. We tested various virus introduction frequency in a naïve herd. We calculated BVD economic impact and vaccination reward. In Charolaise, BVD economic impact was 113€ per cow over 5 years after virus introduction. Irrespective of the vaccine and for a high enough risk of introduction, the yearly expected reward was 0.80€ per invested euro per cow. Vaccination should not be stopped before herd exposure has been decreased. In contrast, the reward was almost nil in Blonde d’Aquitaine and Limousine. This highlights the importance of accounting for herd specificities to assess BVD impact and vaccination efficiency. To guide farmers’ vaccination decisions against BVD, we transformed this model into a French decision support tool.

## 1. Introduction

Bovine viral diarrhoea (BVD) is a worldwide endemic disease of cattle [1], leading to abortions, delayed calving and mortality [2]. Despite the fact control programs have been implemented in many countries [3], BVD is still an issue for farmers [4]. After the virus is introduced in a susceptible herd, an epidemic can occur, which often spontaneously fades out because most infected animals are quickly immunized for life [5]. However, the infection of females in mid-gestation could give rise to the birth of persistently infected (PI) calves [6]. PI animals massively shed the virus for life. They have a shortened lifespan, half dying in their first year [7]. They represent a risk for other animals as well as for herds in contact. On the one hand, the high turnover of animals in cattle herds compromises the establishment of a long-lasting herd immunity. On the other hand, contacts with neighbouring herds on pasture and the purchases of animals potentially carrying the virus lead to regular virus reintroductions [8,9]. When animals spend a long time outdoor on pasture, such as in French cow-calf herds, there is a particular risk of virus introduction through proximity contacts between neighbouring herds at a period when many females are gestating [10].

Several vaccines are available to prevent BVD virus from spreading into non-infected but exposed herds [11]. Vaccination alone is known not to be sufficient to reach eradication at a national scale [11]. In combination with testing, it can lead to eradication but it is generally not economically attractive [12]. However, in the absence of a collective control program, a vaccination performed on a regular basis could help decreasing shedding and new infections in herds exposed to virus introduction. Nevertheless, at this local farm scale, the economic impact of vaccination is not very well known, particularly because BVD spread highly depends on herd characteristics [13], which may impair vaccination efficiency. Such a knowledge would be useful to help farmers decide to vaccinate or not their herd, accounting for herd specificities, such as its size, usage of pasture, calving seasons, etc. In addition, the risk of stopping vaccinating after a few years has not been evaluated yet.

Modelling is a relevant approach to quantify the expected gains or losses (i.e., the reward) associated with vaccinating a herd. Among the many existing models representing BVD virus spread within a cattle herd, a few focused on cow-calf herds [10,14,15,16,17,18]. Early, it has been shown that vaccination coverage has to be very high if PI animals are present [19]. The impact of vaccination in beef herds has been assessed in different farming settings (e.g., in the US: [17], in New Zealand: [18]), with contrasted issues. In New Zealand, for example, killed vaccines are used, which require up to three doses and does not provide a very high foetal protection [18]. In Europe, mostly inactivated vaccines are used, which induce immunity lasting from a few months to a year, and which provide a good foetal protection as long as vaccination is performed before pregnancy [11]. However, how vaccination impact varies in relation with herd characteristics is still barely known, while the practical application of existing tools has been highlighted as a crucial gap of knowledge [4], especially to determine effective ways to control BVD in different production systems. Thus, to produce practical conclusions, models should account for the specificities of the farming systems. This is particularly important for BVD in beef herds because these herds often have a seasonal calving and are exposed to infection at pasture, both being concomitant. In France, several beef farming systems coexist, most with a small to moderate herd size, in which BVD virus circulates and where vaccination impact is hardly quantified.

Our objective was to assess the economic impact of BVD and the reward associated with vaccination against BVD in French beef cow-calf herds, taking into account the characteristics of the farming systems the most commonly encountered in France, as well as the external risk of virus introduction due to proximity contacts and the characteristics of the vaccine. We adapted a mechanistic, partially individual-based, and stochastic epidemiological model of BVD virus spread at herd scale [10] to three farming systems representative of French cow-calf herds, named by their main breed: Charolaise, Blonde d’Aquitaine, and Limousine. We included vaccination in the model to assess strategies defined by the type of protection conferred by the vaccine and the vaccine used. We calculated the yearly and the cumulative economic impact of BVD in these herds, and the expected reward associated with vaccination. Finally, we transformed this into a web decision support tool freely available to French cattle health managers.

## 2. Material and Methods

A stochastic partially individual-based model in discrete time was developed to represent BVD virus spread in a cow-calf herd and to assess the effectiveness of vaccinating breeding females, as usually recommended. A time interval of 7 days enables to properly represent transiently-infected animals in the infection process. The model without vaccination has been previously described [10]. Its main characteristics are briefly recalled hereafter, with a simplification of the computation of the transmission rate during the pasture period. The implementation of the vaccination strategy, new in the model, is described in details.

### 2.1. Herd Dynamics

Animals of the herd can be of seven types: calves from birth until weaning, male and female weaned calves for selling (grassers), heifers younger than 2 years kept for breeding, breeding heifers, cows from the first pregnancy diagnosis until fattening decision, cows from fattening decision until culling, and bulls. The herd management is seasonal, including breeding, calving, indoor and outdoor periods. During the indoor period, we assumed a homogeneous contact structure. The breeding period essentially occurs during the outdoor period. During the outdoor period, animals are on pastures, except remaining grassers which stay indoor until they are sold. Animals are grouped as following: (i) all young heifers, (ii) all breeding heifers plus bulls, and (iii) all calves and cows plus all remaining bulls. The herd size is determined by the number of heifers and cows kept for breeding which is constant for a scenario, so that the farmer can reach his production objective (expected number of animals for sale). In case of unexpected calf mortality or of numerous infertility/abortion events, the farmer purchase replacement calves or pregnant females, respectively. Grassers are sold during the course of the year, at one, two or three different dates. Females detected as empty at the beginning of the indoor period are sold 100 days later. Cows selected for fattening at the beginning of the breeding period are sold after the weaning of their calves.

The model was first parameterized for a Charolaise cow-calf herd structure (Figure 1), the main French beef farming system. Then, it was adapted to represent two other farming systems widespread in France, which main breeds are Limousine and Blonde d’Aquitaine. All the three breeds together constitute more than 60% of the French beef cows. In Limousine and Blonde d’Aquitaine, herds are smaller (respectively 68 and 66 females kept for breeding, against 90 in Charolaise). All the three systems sell their grassers in fall, except Charolaise female grassers sold in spring. The Charolaise system is also the only one to sell some heifers, while the two others keep all their heifers for breeding. In Charolaise, grazing period is the longest, animals on pasture being thus exposed to animals from neighbouring herds for a longer time period. Calf mortality is slightly higher in Charolaise and Blonde d’Aquitaine, and weaning is earlier in Blonde d’Aquitaine (August) compared to Charolaise and Limousine (October). Table 1 provides parameter values for each of the three considered farming systems.

### 2.2. Within-Herd Infection Dynamics and Vaccination

Animals are classified into mutually exclusive health states (Figure 2): susceptible (S), transiently-infected (T), recovered, i.e., immune (R), protected by maternal antibodies (M), persistently infected (P), vaccinated without loss of immunity (V_1_) or vaccinated with progressive loss of immunity (V_2_). The M to S, T to R, and V_2_ to S transitions depend on transition rates Φ_MS_, Φ_TR_, and Φ_VS_ (Table 2). Since the duration of the maternal protection lasts generally 4–6 months [20], we assumed that the M to S transition occurs only in calves.

As classically observed in the field, breeding females (breeding heifers and cows) are vaccinated when they are indoors and just before breeding. This provides the highest level of protection against the risk of birth of P calves. At the start of the breeding period, breeding females in states S and V_2_ go to state V_1_ if vaccinated (ρ_H/C_ = 1). Other health states are not impacted by vaccination. The start of a progressive loss of immunity, i.e., the V_1_ to V_2_ transition, is defined by the duration in V_1_ (Δ_V1_), which is considered as constant and depends on the vaccine. Parameter λ represents the ability of the vaccine to protect animals against infection (reduces the transition to state T), while parameter ω defines the level of protection against vertical transmission if infected (Figure 2). When the vaccine is no longer effective, the animal becomes susceptible again (transition from V_2_ to S, which depends on the mean duration in V_2_ (Δ_V2_), assuming an exponential distribution of the state duration). To account for the diversity of vaccines, parameter values are based on 4 commercial vaccines (Table 3): Bovilis^®^ BVD-MD (Msd Animal Health, Madison, NJ, USA), Bovela^®^ (Boehringer Ingelheim, Ingelheim, Germany), Rispoval^®^ D-BVD (Pfizer, New York, NY, USA), and Mucosiffa^®^ (Merial, Lyon, France). For all vaccines, a protection of at least one year is guaranteed by vaccine producers, thus the duration in V_1_ is assumed to be 52 weeks. The mean duration in V_2_ is unknown, thus we keep it short not to assume a too long vaccine immunity. The level of protection against infection (horizontal transmission) is also unknown. Hence, we assume that infection can occur with the same probability as for susceptible animals, but with a reduced risk of vertical transmission. Infected animals all have a lifelong post-infection immunity. We assess also the opposite case (in a single scenario using Bovilis^®^), i.e., vaccination protecting against infection. In that case, vaccinated animals are infected with a reduced probability compared to susceptible ones, but if infected, the risk of vertical transmission is similar. Most will not be infected, also meaning they will not develop post-infection immunity. All vaccines target BVD virus of type 1, which is the main one circulating in Europe [21]. We assume an average level of protection conferred by the vaccines, without explicitly accounting for the diversity of circulating virus sub-types and its impact on vaccine protection [22].

The force of infection *f* depends on the repartition of the shedding animals (T, P) in the herd. In winter (indoor period), animals are assumed to be all together, thus to be equally in contact with infected animals. The force of infection then is:(1)$$f=\beta^{P}\frac{N^{P}}{N}+\beta^{T}\frac{N^{T}}{N}$$
with N^P^ and N^T^ the total number of P and T animals in the herd, respectively, N the herd size, and β^P^ and β^T^ the transmission rates per day associated with P and T animals, respectively (Table 2).

During the outdoor period, animals are split into separate pastures which do not share any contact. However, the virus can be introduced by contact with infected neighbouring farms. This external risk is modelled through a constant risk *K_ext_*. The force of infection for susceptible animals in pasture *k* is:(2)$$f_{k}=\beta^{P}\frac{N_{k}^{P}}{N_{k}}+\beta^{T}\frac{N_{k}^{T}}{N_{k}}+K_{ext}$$
where $N_{k}^{P}$ and $N_{k}^{T}$ represent the total number of P and T animals in pasture *k*, respectively, and *N_k_* the number of animals in pasture *k*.

Transitions between health states are stochastic, using binomials. Specifically, the number of females changing from vaccinated state V_1_ or V_2_ to infected state T or recovered state R state are calculated as:(3)$$∆N_{VT}=Bin\left(N_{V},\left(1-\omega \right)p\right)$$
(4)$$∆N_{VR}=Bin\left(N_{V},\omega p\right)$$
(5)$$\mathrm{with}\;p=1-e^{-∆t\lambda f}$$
where N_V_ is the number of vaccinated females (either in V_1_ or in V_2_), and ∆t the time step (7 days).

Several consequences including vertical transmission are modelled for infected-pregnant cows, according to the stage of pregnancy at infection time [6]. It is handled by an individual monitoring of breeding females (breeding heifers and cows). Females infected in early (0–41 days), mid-pregnancy (42–150 days), and late (151–285 days) pregnancy are in states R_a_, R_b_ or R_c_, respectively, until calving (Figure 2). Embryonic or foetal deaths are assumed to be highly probable after infection in early or mid-pregnancy (Table 2), after which females join state R. On the contrary, infection occurring in late pregnancy leads to the birth of R calves. When infection occurs in mid-pregnancy, vertical transmission leading to the birth of a P calf is assumed to be highly probable. Vaccinated females are assumed to give birth to calves protected by maternal antibodies (state M).

Finally, calf mortality depends on health state. Non-P calves have a mortality rate μ_Ca_ between birth and weaning, while P animals have a disease-related mortality μ_P_ all their life. P and non-P calves also have a probability of dying at birth, μ_P,bi_ and μ_Ca,bi_ respectively (Table 2).

### 2.3. Model Outputs

The model predicts BVD impact and the vaccination reward in euros per euro invested per cow per year. For each farming system, the meat value, the weight on sale (based on the periods of calving and sale), and the average production cost per kg of meat produced are known (Table 4).

From these prices, weights, production costs and model outputs (i.e., number of animals sold and purchased per group), we calculate the approximate earnings before interest and taxes, which provides an economic measure of the income loss for livestock producers. It accounts for the difference of price between sales and purchases, as well as for the variation in herd size and thus in production cost. The calculation of production cost of animals neither bought nor sold is not necessary since it is considered similar in all situations.

The approximate earnings before interest and taxes are:(6)$$E=\sum_{i\in Sa}N_{i}c_{i}-\sum_{i\in Pu}N_{i}c_{i}-C_{prod}\left(\sum_{i\in Sa}N_{i}k_{i}-\sum_{i\in Pu}N_{i}k_{i}\right)$$
with Sa = {grassers, fattened heifers, culled cows}, Pu = {replacement calves, gestating cows}, C_prod_ the production cost per kg specific to each farming system (Table 4), N_i_ the number of animals in group i, c_i_ the price (in €) of an animal of group i, and k_i_ the weight (in kg) of an animal of group i.

Then, the BVD impact (I) is:(7)$$I=E_{BVDV}-E_{\emptyset BVDV}$$
with E_BVDV_ and E_øBVDV_ the earnings with and without BVD virus spread respectively. Impact thus is negative. However, due to stochastic events, pairs of repetitions with and without BVD spreading cannot be directly compared. We sorted repetitions in order to associate E_BVDV_ and E_øBVDV_ of similar ranking. Nevertheless, impacts still can be positive. In such a case, we considered them as nil. The expected vaccination reward is:(8)$$R=I_{vacc}-C-I_{\emptyset vacc}$$
with C the vaccination cost, and I_vacc_ and I_øvacc_ the BVD impacts with and without vaccination, respectively. The vaccination cost depends on the vaccine characteristics (price and number of doses required; Table 3), as well as on the number of females vaccinated, i.e., the number of breeding heifers and cows which is constant. An average cost is estimated based on the number of doses administered, knowing that for a first vaccination (for heifers) the doses can be higher:(9)$$C=\frac{(N_{H}+N_{C})\rho_{H/C}n_{1}c_{D}+(N_{H}n_{1}+N_{C}n_{2})\rho_{H/C}c_{D}\left(y-1\right)}{y}$$
with N_H_ and N_C_ the number of breeding heifers and cows, respectively, ρ_H/C_ the boolean indicating if there is vaccination (Table 3), n_1_ and n_2_ the number of vaccine doses for the first shot and for the following ones respectively, c_D_ the cost of a vaccine dose, and y the number of years of vaccination. The first year, all breeding females are assumed to be at their first shot. Finally, the euros per euro invested per cow per year is:(10)$$Ei=\frac{R_{cumul}}{N_{y}\;C}$$
with *R_cumul_* the cumulated reward over the simulated years, N_y_ the number of simulated years, and C the average vaccination cost per year. This output has the advantage of combining the information associated with disease losses, vaccination cost, and reward.

The model was developed in C++ and Python, and is available under Apache 2 License here: https://forgemia.inra.fr/sandie.arnoux/bvd-within-herd-model-public (accessed on 4 October 2020). It has been adapted as a tool for French animal health managers (in French), EvalBVD (https://mihmestools.eu/evalbvd/home/ (accessed on 4 October 2020)). It corresponds to a freely available web interface, where all the parameters related to the farming system and the vaccination strategy can be set to represent a specific situation.

### 2.4. Simulation Settings

The expected number of P calf births in a BVD-free herd exposed to a given external risk *K_ext_* is:
(11)$$\left(1-{\left(1-K_{ext}\right)}^{\frac{W}{2}}\right)\left(1-\alpha_{Rb}\right)\eta_{P}N_{F}{∆}_{t}$$
where K_ext_ is the external risk, W the number of weeks on pasture, αRb the abortion rate due to infection mid-pregnancy, η_P_ the probability of giving birth to a calf in state P after infection in mid-pregnancy and if no abortion, N_F_ the number of breeding females, and Δt the time interval. (1−K_ext_)^(W/2) is the probability not to be infected in mid-gestation on pasture. Using Equation (11), 0.4, 1.9, and 7.7 P calf births are expected if the BVD virus is introduced in a naïve Charolaise herd (W = 33 weeks, N_F_ = 90) exposed to K_ext_ = 0.00005, 0.00025, and 0.001, respectively.

A reference scenario considers a BVD-free Charolaise herd consistently exposed outdoor to an external risk K_ext_ of 0.00025 over an 8-year period. We compared results without and with vaccination of all breeding heifers and cows with the vaccine Bovilis^®^.

We also tested complementary scenarios, modifying part of the settings of this reference scenario. First, we compared the efficiency of four commercial vaccines (Table 3) under a large range of K_ext_ values (from 0 to 0.001). Second, we compared the three farming systems (Charolaise, Limousine and Blonde d’Aquitaine) under two K_ext_ values (one moderate 0.00025 and one high 0.001). To reduce computational costs and because the largest losses due to BVD in beef cow-calf herds are known to occur in the first few years after virus introduction, we simulated 5 years for all these scenarios.

Finally, we assessed the effect of stopping vaccination in a Charolaise herd, using Bovilis^®^, and for a large range of K_ext_ values (from 0 to 0.001). The scenario was explored over 20 years: vaccination was implemented the first 6 years, and then stopped. We considered two contrasted protections conferred to vaccinated animals: no infection (λ = 1 and ω = 0) *versus* no vertical transmission if infected (λ = 0 and ω = 1). For each scenario, we ran 1000 repetitions to ensure model output stability.

### 2.5. Sensitivity Analysis

Vaccination reward is computed from BVD impact and vaccination cost. Therefore, reward and impact are highly correlated. Thus, we limited the sensitivity analysis of the model to BVD impact. We varied simultaneously all epidemiological parameters by plus or minus 25% of their nominal values. A complete factorial design [33] was used to estimate the sensitivity indices related to the principal effect and to interactions between parameters for each of the parameter tested, μ_P,bi_, μ_P_, β^P^, β^T^, Φ_MS_, Φ_TR_, α_Ra_, α_Rb_ and η_P_. To limit the number of simulations, closely related parameters were grouped (β^P^ and β^T^, Φ_MS_ and Φ_TR_, α_Rb_ and η_P_) and thus varied in the same direction at the same time. A total of 729 scenarios were run, with 1000 repetitions per scenario.

## 3. Results

### 3.1. BVD Impact and Vaccination Reward in Exposed Beef Cow-Calf Herds of Charolaise System

In a naive cow-calf herd in the Charolaise breeding system, the greatest BVD impact occurred on average 3 years after the herd began to be exposed to the virus due to close contact with infected farms (Figure 3). Over five and eight years, it reached 113€ and 163€ per cow, respectively. If vaccination started immediately (i.e., when exposure started), the associated reward was positive from years 3 to 5, when the BVD impact was high. The rest of the time, the vaccination cost per year overpassed the BVD impact per year, either because consequences due to infection were not already visible (year 1 and 2) or because of herd immunity which has developed after the natural infection has occurred (year 6 and after). Nevertheless, the median reward per year over the whole period was positive. It was 0.80€ per invested euro per vaccinated animal per year over the first five years (percentiles 25/75 being 0.21/1.40€).

### 3.2. Influence of the Vaccine Used and of the External Risk of Virus Introduction

Irrespective of the vaccine used, vaccination was beneficial when the external risk exceeded the threshold of 7 × 10^−5^, which corresponds to one new P calf born on average every two years in a fully susceptible herd (Figure 4). Then, for an external risk of 0.00025, the reward over the first five years increased up to 0.85€ per invested euro per year and per vaccinated animal. An external risk of 0.00025 corresponds to two persistently infected calves born on average over a pasture period in an exposed naive herd. A plateau was reached: the reward did not increase further for higher values of the external risk. It has to be noted the extreme variability of the reward for a given external risk: it varied from −3€ to +4€ per invested euro if the external risk was at least moderate, the variations occurring among repetitions of a given scenario, thus related to the model stochasticity and the occurrence of quite rare events such as the birth of a P calf, or on the contrary its death quickly after birth.

### 3.3. Comparison of Three French Farming Systems

While vaccination is not beneficial for all external risks, it is also not beneficial for all farming systems (Figure 5). The reward highly varied for a given situation (farming system x external risk), being positive in more than 75% of the cases in Charolaise if the external risk was higher than 0.00025. For the two other farming systems, Limousine and Blonde d’Aquitaine, the reward was predicted to be positive in half the cases and negative in the other half, even for large external risks. As a result, and in contrast to Charolaise, the median reward was almost nil in Blonde d’Aquitaine and Limousine (percentiles 25/75 being −0.80/0.35€, and −0.72/0.37€, respectively).

### 3.4. When to Stop Vaccinating?

In Charolaise, for a moderate external risk and if this risk has not been decreased, stopping vaccinating was not relevant even if the reward was predicted to be negative after six years (Figure 6). Indeed, a few years after stopping vaccinating breeding females, herd immunity becomes sufficiently low and the same situation occurs again (brown and orange curves in Figure 6).

Similar results were obtained when assuming the vaccine fully protects against infection (animals becoming susceptible again once vaccine immunity has faded out) or when assuming the vaccine only protects against vertical transmission (thus infection can occur–and animals then become immune for life-but no P calf can be born). As expected, the second option reduced the BVD impact after vaccination has stopped, as more animals are immune in that situation, but the reward difference between the two types of protection was small.

### 3.5. Sensitivity of BVD Impact and Vaccination Reward to Epidemiological Parameters

Varying the epidemiological parameters by 25% of their nominal values in a beef cow-calf herd of Charolaise breeding system exposed to a moderate external risk of virus introduction (K_ext_ = 0.00025) induced a variation of the median BVD impact, which ranged between −136 and −87€ per breeding female. Most of this variation (70%) was explained by the consequences of infection occurring during female pregnancy, especially during its early stage, the infection-related abortion rate in early pregnancy explaining half the variance of this model output (Figure 7). More surprisingly, the mortality rate of P animals and the transmission rates barely contributed to the variation of the BVD impact. No interaction between parameters contributed to explain output variations.

## 4. Discussion

The results of our study highlight the importance of accounting for herd specificities to estimate BVD impact on beef cow-calf herd production and to assess vaccination efficiency to reduce this impact locally at herd scale. While BVD has an impact in Charolaise breeding system, irrespective of the commercial vaccine used and as long as the external risk was high enough, the median reward was almost nil in Blonde d’Aquitaine and Limousine breeding systems. This emphasises the large influence the farming system can have on BVD impact and on vaccination added value. French beef cow-calf seasonal herds are quite representative of other farming systems used in Europe (e.g., in Ireland: [34]). Although the three farming systems studied here could be considered as having quite similar management, the differences in the estimated BVD impact clearly indicate the need for looking specifically to BVD impact and vaccination reward for each farming system and not to extrapolate from results obtained with other farm settings. This strengthens the need for practical decision support tools to provide farmers and their advisers with estimations of BVD impact, and to guide farmers’ decision on BVD vaccination. Accounting for the precise situation of the farm in terms of exposure to BVD virus introduction and of herd management is crucial, such as done in the EvalBVD tool.

When vaccination is effective, we showed that it is better not to stop vaccinating as long as the risk of BVD virus introduction due to proximity contacts with neighbouring herds has not been decreased. This is an important and practical finding for farmers, as the decision to stop controlling can be as hard to take as the decision to start. Indeed, BVD impact seems better explained by abortions than by the death of PI calves, as also found in the US [17]. Herd immunity does not last long enough to protect the herd over long period if vaccination is stopped. This is partly because of the herd renewal rate, partly because of the rather short duration of the individual immunity conferred by vaccination.

It emphasises the relevance of regional collective control strategies aiming at reducing BVD exposure at a larger scale than the single herd, using vaccination on such larger scale combined with the test-and-cull of persistently infected animals [35]. However, these collective strategies sometimes cannot be associated with vaccination, when serological tests are performed, as inactivated vaccines interfere with these tests [36]. In any case, it remains difficult to estimate the risk of BVD introduction by proximity contacts, and thus to assess its variation over time and with control implementation. This might impair the assessment of the efficiency of within-herd control measures, such as vaccination. Comparing costs and benefits of vaccination *versus* PCR testing to identify and remove persistently infected animals, analyses carried out in the Swiss eradication context show a return on investment of four years with individual testing [37]. Combining compulsory vaccination and testing also appears as efficient [12]. However, vaccination alone does not enable achieving eradication. It induces permanent costs that at best reduces losses below the cost of vaccination. In contrast, eradication, once reached, reduces control costs to that of herd serological surveillance, which is very moderate compared to disease costs and to vaccination costs. It should also be noted that PCR costs have largely decreased in recent years, e.g., being now in France in the order of 3–4€ per animal tested.

The present study considered only the BVD virus of type 1, the main one circulating in France [38]. Also, we neglected the diversity in co-circulating strains, while virus subtypes are numerous [39,40] and vaccines does not protect against all of them [22]. This may further impair vaccination efficiency. It is an interesting modelling perspective: a multi-strain model would for example enable to assess the reward expected from using a multivalent vaccine and cross-protection against diverse virus sub-types.

We have highlighted that vaccination reward highly depends on the epidemiological situation and the farming system, but also varies within a given situation. The reward heterogeneity possibly explains why farmers’ decisions with regards to BVD control are so heterogeneous in the field. To better account for the strategic behaviour of farmers and assess collective control scheme at large scale, there is a need for models combining farmers’ decisions and the dynamics of the epidemiological situation [41].

## Figures and Tables

**Figure 1 vaccines-09-01137-f001:**
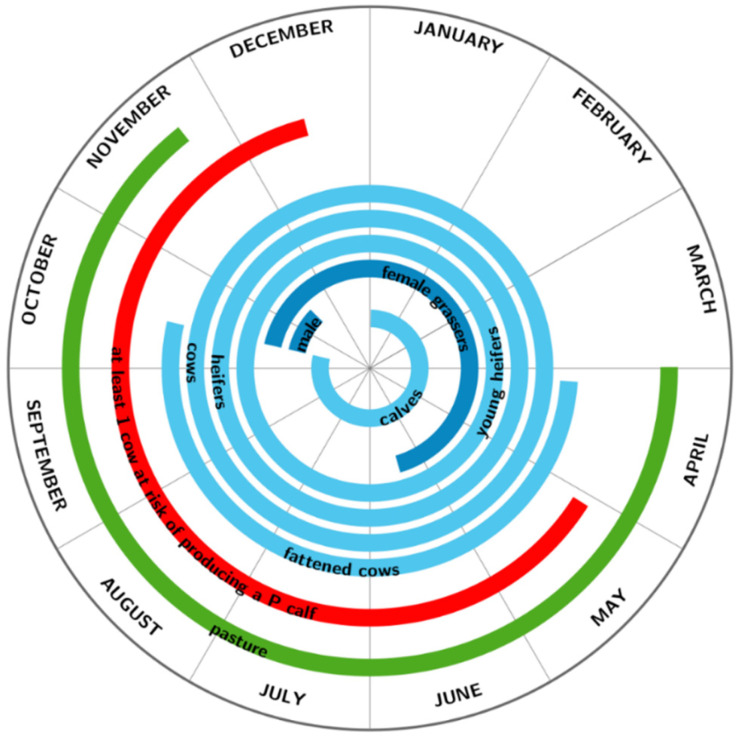
Calendar of the Charolaise farming system. The periods where each of the 7 animal groups are present in a cow-calf herd across a year are shown in blue. With a breeding period from end of March to end of July, the risk period is from start of May to mid-December, where cows can be in mid-pregnancy and vertical transmission is then highly probable if infection occurs. Animals can be in contact with infected neighbouring farms during the pasture period. Only male and female grassers are kept indoor.

**Figure 2 vaccines-09-01137-f002:**
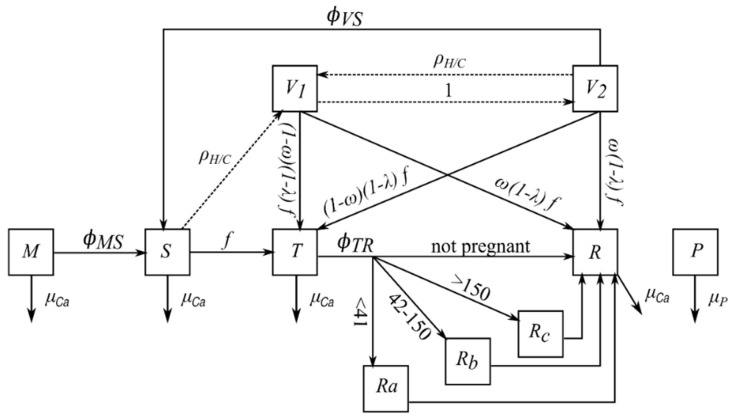
Transitions between health states. M: protected by maternal antibodies; S: susceptible; T: transiently infected; P: persistently infected; R: immune; R_a_, R_b_, R_c_: immune and pregnant, infected respectively in early, mid-, and late pregnancy; V_1_: vaccinated without loss of immunity; V_2_: vaccinated with progressive loss of immunity. Symbols ρ_H/C_, ω and λ stand respectively for the vaccination of breeding heifers and cows (Boolean), the protection conferred by the vaccine against vertical transmission, and the protection conferred by the vaccine against infection. Health state M, mortality rate μ_Ca_, and transition rate Ф_MS_ only concern calves. Health states R_a_, R_b_, R_c_, V_1_ and V_2_ only concern breeding heifers and cows. Transitions from S to V_1_ and from V_2_ to V_1_ occur once a year at the beginning of the breeding period (ρ_H/C_ = 1). Transition from V_1_ to V_2_ depends on the duration in state V_1_ (Δ_V1_). The force of infection f is derived from Equations (1) and (2), other parameters are given in Table 2 and Table 3.

**Figure 3 vaccines-09-01137-f003:**
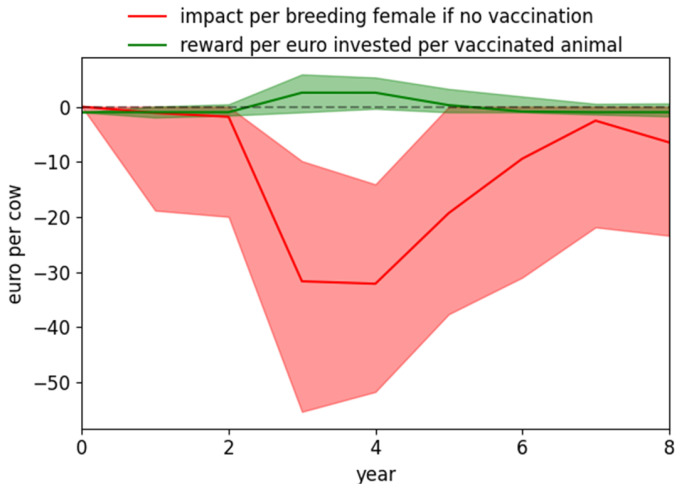
BVD impact and reward after vaccination (in € per breeding female). A beef cow-calf herd of Charolaise breeding system is exposed to a moderate external risk of virus introduction (K_ext_ = 0.00025). The Bovilis^®^ vaccine is provided yearly to all breeding females before the start of the breeding period. The envelop shows the percentiles 25 and 75 of the 1000 stochastic repetitions for both scenarios.

**Figure 4 vaccines-09-01137-f004:**
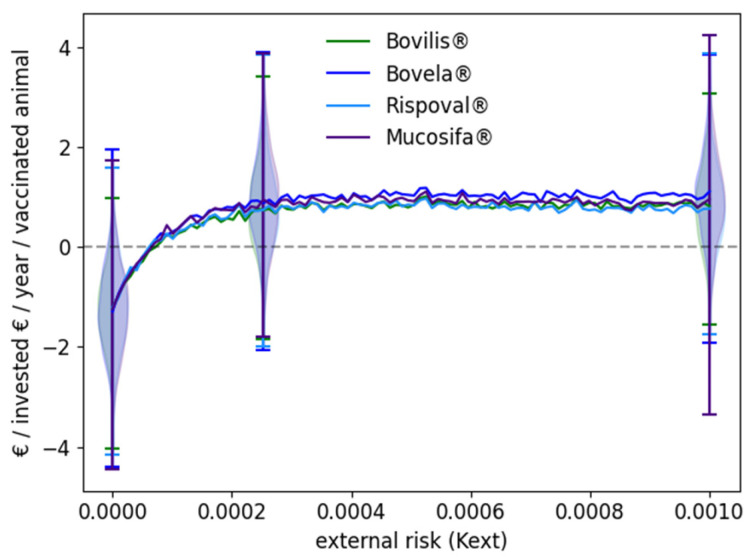
Reward after vaccination against BVD according to the vaccine used and the external risk of virus introduction. A beef cow-calf herd of Charolaise breeding system is considered. All breeding females are vaccinated yearly before the start of the breeding period using one of four commercial vaccines (see Table 1 for details on vaccines). The curves show the median over 1000 stochastic repetitions of the euros obtained per euro invested per year per vaccinated animal over 5 years according to the external risk. The violin plots show the values for the 1000 stochastic repetitions for three K_ext_: 0, 0.00025, 0.001. Each side of the violin plots shows two overlapping scenarios: Bovilis^®^ and Bovela^®^ on the left; Rispoval^®^ and Mucosiffa^®^ on the right.

**Figure 5 vaccines-09-01137-f005:**
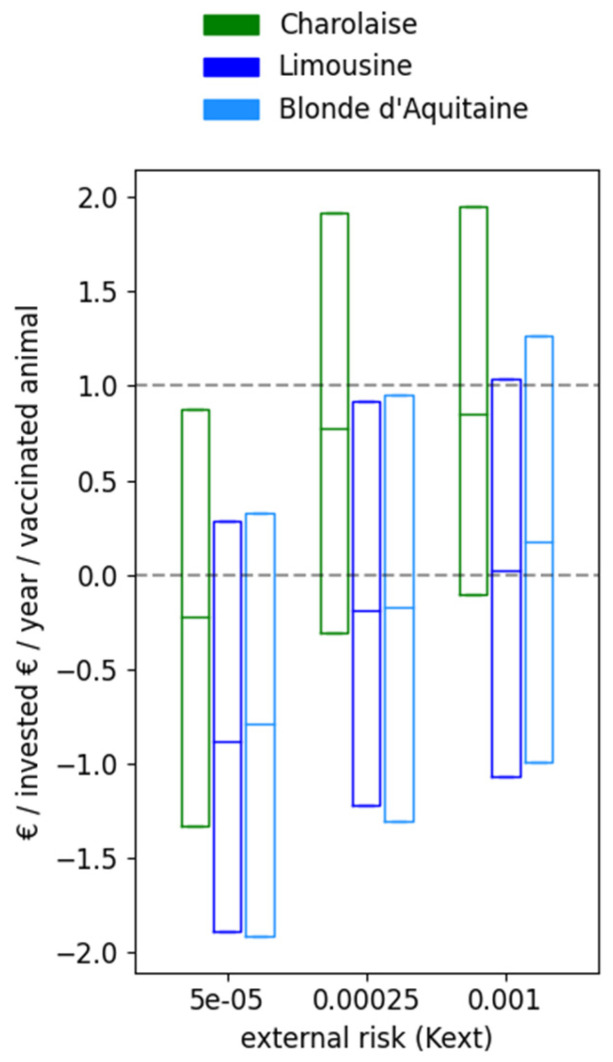
Return on investment after vaccination against BVD according to the farming system and the external risk of virus introduction. All breeding females are vaccinated yearly before the start of the breeding period using Bovilis^®^. Three farming systems are compared: Charolaise, Limousine and Blonde d’Aquitaine. The percentiles 10, 50 and 90 of the euros per euro invested per year per vaccinated animal over 5 years are shown from 1000 stochastic repetitions.

**Figure 6 vaccines-09-01137-f006:**
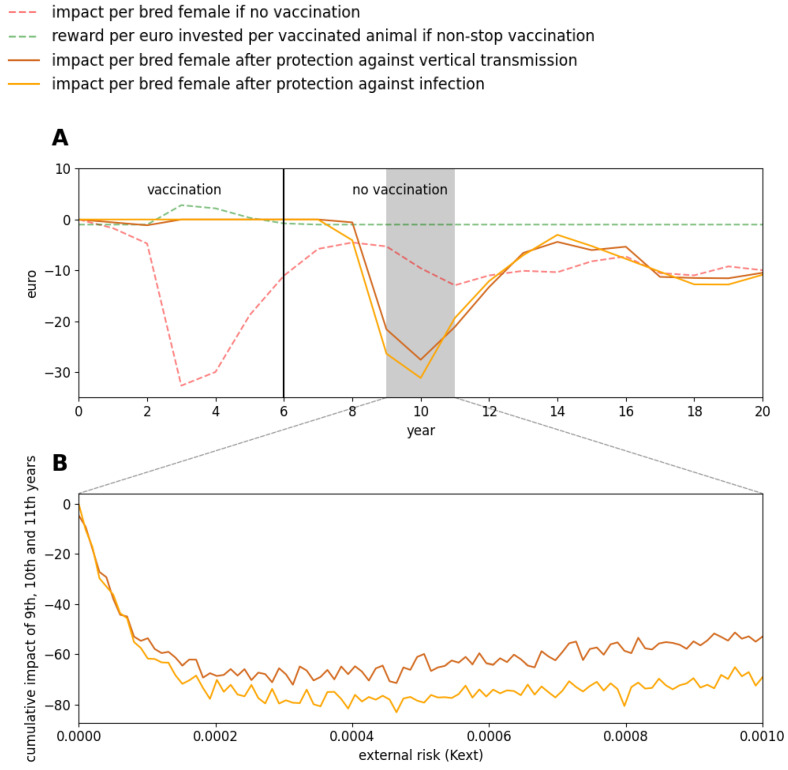
BVD impact and vaccination reward according to the type of protection conferred by the vaccine. A beef cow-calf herd of the Charolaise breeding system is considered. In scenarios with vaccination, all breeding females are vaccinated yearly with Bovilis^®^ before the start of the breeding period. Vaccination is stopped after 6 years and 2 types of protection are tested: against vertical transmission and against infection. (**A**): BVD impact per breeding female and reward per euro invested per vaccinated animal when assuming a moderate external risk (K_ext_ = 0.00025); (**B**): Cumulative BVD impact per breeding female over three years (years 9, 10 and 11) according to the external risk. In all cases, the median over 1000 stochastic repetitions is shown.

**Figure 7 vaccines-09-01137-f007:**
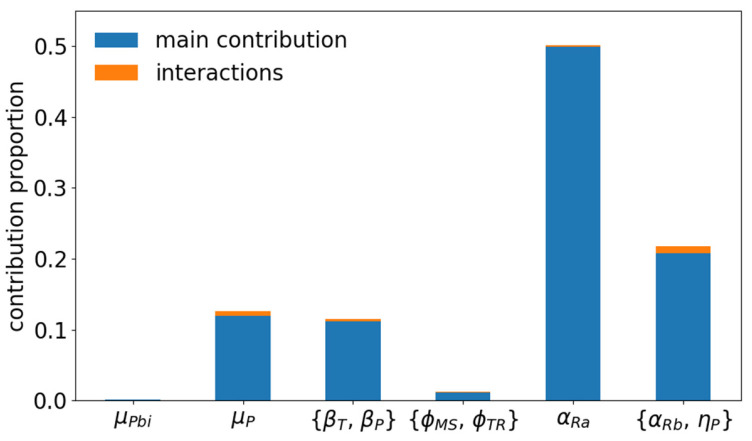
Sensitivity of BVD impact to variations in epidemiological parameters (±25%), using a complete factorial design. A beef cow-calf herd of the Charolaise breeding system is considered. Parameters are from left to right: mortality of P calves at birth, mortality of all P animals, transmission rates, transition rates between health states other than infection, abortion rate in early pregnancy, other consequences of infection during pregnancy.

**Table 1 vaccines-09-01137-t001:** Values of model parameters for the three farming systems.

Parameters	Charolaise	Limousine	Blonde d’Aquitaine
Number of cows kept for breeding	68	56	53
Number of heifers kept for breeding	22	12	13
Probability of infertility of cows	0.061	0.048	0.080
Probability of infertility of heifers	0.039	0.026	0.058
Probability of having twins	0.023	0.004	0.022
Probability of calf mortality before weaning	0.08	0.05	0.07
Weaning date	14 October	1 October	29 August
Breeding period	22 March–19 July	4 February–4 June	19 February–19 July
Pasture period	1 April–22 November	1 April–14 November	10 April–9 November
Pasture length	237 days	229 days	215 days
Sale period of male grassers	15 November (+/−3 weeks)	15 November (+/−3 weeks)	
Sale period of female grassers	15 June (+/−3 weeks)		
Number of heifers sold at breeding	2	0	0

**Table 2 vaccines-09-01137-t002:** Epidemiological parameters.

Scheme	Definition	Value	Reference
μ_P,bi_	Probability of mortality at birth of P calves	0.0667	[7,23]
μ_Ca,bi_	Probability of mortality at birth of non-P calves	0.02	
μ_P_	Mortality of P animals per day	0.0019	[7,23]
μ_Ca_	Mortality of non-P calves per day	0.000326	
Ф_MS_	Transition rate from state M to state S per day	0.00667	[24]
Ф_TR_	Transition rate from state T to state R per day	0.2	[25]
Ф_VS_	Transition rate from state V_2_ to state S per day	1/ Δ_V2_	
β^T^	Transmission rate for T animals	0.03	[26,27]
β^P^	Transmission rate for P animals	0.5	[26,28]
α_Ra_	Abortion rate due to infection early pregnancy	0.8	[6,25,29]
α_Rb_	Abortion rate due to infection mid-pregnancy	0.2	[6,30,31]
η_P_ η_M_ = η_R_	Probability of giving birth to a calf in state P, M, or R if infection in mid-pregnancy and no abortion	0.9375 0.03125	[6,28,30,31,32]
K_ext_	External risk of virus introduction during pasture	0–0.001	

**Table 3 vaccines-09-01137-t003:** Vaccine parameters.

Scheme	Definition	Bovilis^®^	Bovela^®^	Rispoval^®^	Mucosiffa^®^
ρ_H/C_	Boolean	1 if vaccination, 0 otherwise			
λ	Probability of protection against infection ^£^	0/1 *	0	0	0
ω	Probability of protection against vertical transmission ^§^	1/0 *	0.985	0.9	0.9
Δ_V1_	Duration in V_1_ state (weeks) ^§^	52	52	52	52
Δ_V2_	Duration in V_2_ state (weeks) ^£^	8	8	8	8
c_D_	Dose cost (euro) ^§^	4.59	5.80	4.33	5.72
n_1_	Number of vaccine doses for the first shot ^§^	2	1	2	1
n_2_	Number of vaccine doses for the second shot ^§^	1	1	1	1

***** According to the type of protection considered. ^§^ Commercial information (vaccine producers, vet costs, etc.). ^£^ Assumed.

**Table 4 vaccines-09-01137-t004:** Weight and price of sold and purchased animals. Parameter values depend on the farming systems, based on data from Inosys Réseaux d’élevage (2014, https://idele.fr/detail-dossier/cas-types-bovins-allaitants (accessed on 4 October 2020)).

Parameters		Charolaise		Limousine		Blonde d’Aquitaine	
		€	Kg	€	Kg	€	Kg
Price and weight per animal type	Replacement calf	352	47	352	42	352	46
	Gestating cow	1689	747	1621	682	1828	814
	Male grasser	1051	420	966	315	915	260
	Female grasser (sold in fall)	-	-	793	300	724	240
	Female grasser (sold in spring)	985	400	-	-	-	-
	Fattened heifer	1330	630	-	-	-	-
	Culled cow	1337	700	1570	691	1710	798
C_prod_	Production cost per kg	0.754	-	0.652	-	0.775	-

## Data Availability

The code is available under Apache 2 License at https://forgemia.inra.fr/sandie.arnoux/bvd-within-herd-model-public (accessed on 4 October 2020).

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
