# Peer review of "To Vaccinate or Not: Impact of Bovine Viral Diarrhoea in French Cow-Calf Herds"

_vaccines, 2021, doi:10.3390/vaccines9101137_

Round 1
Reviewer 1 Report
This study is in line - and a logical consequence of previously stated questions - with a surprisingly large number of modelling attempts that have been put forward about the control of BVDV. The merit of the here described model is the calculation of individual price tags for BVD control via vaccination, depending on the type and brand of vaccine and the herd management of typical cattle farming in France. Interestingly, vaccination yields positive cost benefits only for the Charolais farming while for other types this is less apparent or even absent.
While BVD control by vaccination and/or eradication has been discussed in the past 30 years with religious zeal (and sometimes national chauvinism), the cost benefit was often a matter of belief. Retrospective models in the last 10 years have provided sound numbers as the (taxpayers) efforts for eradication are high and vaccination insufficient for eradication.
My only recommendation for this excellent paper would be to compare the cost benefits of vaccination with published about eradication. In a setting of large and extensive cattle farming (US, Brasil) vaccination is the only option to reduce clinical losses but is this true for Europe? Do the costs of testing calves by PCR (after all the best way to find PIs) exceed vaccination?
The discussion puts further complications such as strain dependent virulence into perspective of future studies. I would not recommend this efforts as for most BVDV strains data about virulence are absent. Clinical data from the field are not controlled at all and I'm not aware of systematic challenge experiments. The few strains that have been characterised, mostly in combination with vaccination studies are extinct or have evolved into something else.
Author Response
Please find our answers in the attached file.

Reviewer 2 Report
The authors estimated economic impact of BVDV vaccination in different French cow-calf heard. Interestingly, this impact depends on farming system used in the heard.
Minor comments:
line 102 - correct typo in this line;
line 109 - change "as follow" to "as following"
line 198-199 - change format of table 3. The table should be on one page.
Author Response

(The authors gave the same response as above.)
